# The Hox Gene *egl-5* Acts as a Terminal Selector for VD13 Development via Wnt Signaling

**DOI:** 10.3390/jdb8010005

**Published:** 2020-03-03

**Authors:** Meagan Kurland, Bryn O’Meara, Dana K. Tucker, Brian D. Ackley

**Affiliations:** 1Department of Molecular Biosciences, The University of Kansas, Lawrence, KS 66045, USA; meagankurland@ku.edu (M.K.); bryn.omeara@ku.edu (B.O.); 2Department of Biology, The University of Central Missouri, Warrensburg, MO 64093, USA; dtucker@ucmo.edu

**Keywords:** Wnt signaling, Hox genes, neuron specification, axon outgrowth

## Abstract

Nervous systems are comprised of diverse cell types that differ functionally and morphologically. During development, extrinsic signals, e.g., growth factors, can activate intrinsic programs, usually orchestrated by networks of transcription factors. Within that network, transcription factors that drive the specification of features specific to a limited number of cells are often referred to as terminal selectors. While we still have an incomplete view of how individual neurons within organisms become specified, reporters limited to a subset of neurons in a nervous system can facilitate the discovery of cell specification programs. We have identified a fluorescent reporter that labels VD13, the most posterior of the 19 inhibitory GABA (γ-amino butyric acid)-ergic motorneurons, and two additional neurons, LUAL and LUAR. Loss of function in multiple Wnt signaling genes resulted in an incompletely penetrant loss of the marker, selectively in VD13, but not the LUAs, even though other aspects of GABAergic specification in VD13 were normal. The posterior Hox gene, *egl-5*, was necessary for expression of our marker in VD13, and ectopic expression of *egl-5* in more anterior GABAergic neurons induced expression of the marker. These results suggest *egl-5* is a terminal selector of VD13, subsequent to GABAergic specification.

## 1. Introduction

Nervous systems are organized into functional units comprised of diverse cells. Knowing a cell’s identity enables us to trace its lineage, describe the architecture and connectivity of networks and elucidate the physiology of organismal behaviors. The original attempt to systematically categorize neurons was by Ramon y Cajal, who beautifully illustrated conserved, distinguishing neuronal morphologies in different animals [1]. Today, neuroscientists categorize neurons using anatomical, morphological, molecular or functional criteria, e.g., hippocampal, pyramidal, tyrosine hydroxylase or motor neuron, etc., are different, but informative, labels for specific classes of neurons.

During development, neurons acquire differential features through iterative rounds of specification in a hierarchical fashion. These rounds are driven by both extrinsic cues, which provide spatial information to the cells, and intrinsic transcriptional programs orchestrated by transcription factors and terminal selectors. For example, motorneurons (MNs) within the vertebrate spinal cord are initially specified by Olig2 [2,3], after which they organize into discrete motor columns along the anterior-posterior axis [4,5], and then into subtypes or motor pools, which correspond to specific muscle regions within the limbs. Many different transcription factors exhibit selectivity for the different sub-divisions along the A/P axis and within the functional regions (for review see [6,7]). Of these, Hox genes play a critical role in multiple events of MN specification along the A/P axis (for review see [8]).

*Caenorhabditis elegans* (*C. elegans*) have both excitatory (cholinergic) and inhibitory (γ-amino butyric acid/GABA-ergic) MNs. These MNs collaborate to control body wall muscles to produce smooth sinusoidal locomotion. MNs can be categorized based on when they form, the specific body regions innervated and their function. The loss of GABAergic MN function results in a “shrinker” phenotype, which enabled screens for genes that regulate GABAergic specification and function [9]. Two of those genes—*unc-25*, which encodes glutamatergic acid decarboxylase, the final biosynthetic step in GABA synthesis, and *unc-47*, which encodes the vesicular GABA transporter—have provided differentiation markers that have also been used to further understand GABAergic specification [10,11].

There are several genes that have been identified to play a role in GABAergic specification. *cdn-1* is necessary for the proper specification and differentiation of both GABAergic and cholinergic MNs [12]. *unc-30* functions downstream of *cdn-1* to specify the fate of the D-type GABAergic MNs [13,14]. The Aristaless homolog, *arl-1, unc-62* and *unc-55* repress DD-like fates and promote the specification of VD fates [15,16,17,18]. Together, these provide a small subset of the genes that are involved in the specification of GABAergic MNs in *C. elegans*.

Here, we report a new marker specific for the most posterior GABAergic D-type MN, VD13, and a pair of bilaterally symmetrical neurons that we have tentatively identified as the LUA neurons, LUAL and LUAR. Using this marker, we observed developmental defects in the morphology of VD13 in animals with mutations in Wnt signaling genes, including animals where expression of the marker was lost from VD13, but not the LUAs. Expression was completely lost in VD13 in animals lacking the β-catenin, *bar-1*, and the Hox gene, *egl-5*. Additionally, we found redundant function in regulating expression for two disheveleds (*dsh-1* and *mig-5*) and two Wnt ligands (*lin-44* and *egl-20*). Our results indicate that Wnt signaling promotes *egl-5* to function as a terminal selector for the VD13 fate.

## 2. Materials and Methods

### 2.1. Strains and Genetics

N2 (var. Bristol) was used as the wild-type reference strain in all experiments. Strains were maintained at 18–22 °C, using standard maintenance techniques as described [19]. Alleles used in this report include LGI: *lin-44(n1792), lin-17(n671);* LGII: *mig-5(rh94), mig-5(tm2639), dsh-1(ok1445);* LGIII: *egl-5(n945);* LGIV: *egl-20(gk453010), egl-20(lq42)* [20]. The following integrated strains were used: LGII: *juIs76* [*Punc-25::gfp*]; LGV: *wgIs54* [*Pegl-5::egl-5::gfp*]; LGX: *lhIs97* [P*plx-2*::*mCherry*]. The following extrachromosomal arrays were generated and used in this report: *lhEx609, lhEx610* [P*plx-2*::*rfp*], *lhEx555* [P*unc-25*::*egl-5 cDNA*]. The *lhEx555* array was generated by injecting pEVL479 (P*unc-25::egl-5*) into wild-type animals at 2 ng/μL, along with pBA183 (P*myo-2::mCherry*) at 5 ng/μL. The *lhEx609, lhEx610* arrays were generated by injecting pEVL551 into wild-type animals at 5 ng/μL along with P*str-1::gfp* at 20 ng/μL. *lhEx609* was integrated into the genome using ultraviolet light and tetramethylpsoralen (UV/TMP), to generate *lhIs97*, and backcrossed to wild-type six times prior to use. 

### 2.2. Plasmid Construction

pBA102 (P*plx-2::cfp::unc-54 3′utr*) contains a segment of the *plx-2* promoter (nucleotides -3863 to -871 with respect to the start of translation). pBA102 was digested with *Bam*HI and religated to create a shortened *plx-2* promoter (contains nucleotides -1089 to -871, with respect the start of translation, to create pEVL530 (P*plx-2*::*cfp*::*unc-54* 3′UTR). pEVL387 (P*unc-25*::*mCherry*::*unc-43 3′UTR*) was digested using *Xma*I and *Spe*I and ligated to pEVL550 digested with *Xma*I and *Spe*I to generate pEVL531 (P*plx-2::mCherry*::*unc-43 3′ utr*). 

The *egl-5* cDNA was amplified from total RNA that had been converted to cDNA using random hexamer oligos and superscript III reverse transcriptase (Life Technologies). The cDNA was amplified using the following primers, *egl-5F1* 5′-ttggaaagcagtgagagtgag-3′ and *egl-5 R1* 5′-ggagggatcattgagaaacttgag-3′ and inserted into a vector with the *unc-25* promoter and *unc-54* 3′UTR using LR Clonase (Life Technologies). All vector and plasmid sequences are available upon request.

### 2.3. Fluorescence Microscopy

The complete set of GABAergic MNs was visualized using *juIs76*, while VD13 and the LUAs were scored using *lhIs97*. Scoring and imaging were done using an Olympus FV1000 laser-scanning confocal microscope with the Fluoview software. Animals with posterior neurites (Pdns) or spurious *lhIs97* expression in DD6 were not included in the shape analysis of VD13. Images were exported to ImageJ to be rotated and/or cropped for presentation. The ImageJ Image Calculator function was used to make the image of the intersection between *juIs76* and *lhIs97*.

### 2.4. Statistics

Fisher’s exact test was used to evaluate statistical significance between genotypes, and calculated with Prism GraphPad (5.0) or GraphPad QuickCalcs (http://www.graphpad.com/quickcalcs/). All genotypes were scored on a minimum of two different days, and the results averaged between scoring sessions. We set a threshold of *P* < 0.005 to determine significance to account for multiple testing. The standard deviation of the population was used to calculate error bars for the reported VD13(+) expression in the varying genotypes and was calculated using Microsoft Excel 2016.

### 2.5. Data Availability

All strains and plasmids presented are available upon request. 

## 3. Results

### 3.1. Isolation of a VD13-Selective Marker

There are 19 GABAergic primary motorneurons (MNs) in the *C. elegans* locomotor circuit. The six dorsal D-type (DD) neurons form during embryogenesis [21]. These neurons are presynaptic to the ventral body wall muscles during the first larval stage (L1) [22]. However, during L1, the 13 ventral D-type (VD) neurons begin forming. The most anterior neuron, VD1, forms first, while the most posterior, VD13, forms last. During the L1 stage the DD neurons remodel to innervate the dorsal muscles, as VD neurons innervate ventral muscles [22,23,24].

Using GFP (green fluorescent protein) reporters under the control of GABA-specific promoters, several labs have investigated the development of the GABAergic MNs. Briefly, the D-type GABAergic MNs share a common morphology, a sideways H shape, with the cell body on the ventral midline. During development, the cell body extends an axonal-like process anteriorly, from which a commissural process forms, and bifurcates at the dorsal nerve cord, where anteriorly and posteriorly directed processes extend [25]. In general, the processes reach to the next D-type neuron of the same class to ultimately become tiled along the body (Figure 1). 

We serendipitously isolated a reporter transgene active in VD13 and the bilaterally symmetrical LUA neurons. This transgene uses a fragment of DNA upstream of the *plx-2*/Plexin gene [26]. We created an integrated transgene, *lhIs97*, with this promoter fused to mCherry (Figure 1). In conjunction with a pan-GABAergic MN marker *juIs76* (P*unc-25::*GFP) [27], mCherry and GFP overlapped uniquely in VD13 (Figure 1). The cell-selective expression of the *lhIs97* transgene suggested VD13 was undergoing developmental events separate from other GABAergic MNs.

In wild-type animals, we found that, rather than being “H” shaped, VD13 was most frequently “C” shaped (82 ± 1% of animals observed) (Table 1, Figure 2). In approximately 9% of animals, the cells had either a T shape where the cell extended an additional, anteriorly directed process in the dorsal nerve cord, or only an anterior process (“P” shaped). Interestingly, we also found that in 9% of animals no VD13 commissural process was visible (“N” shaped), or it formed, but failed to reach the dorsal nerve cord (“O” shaped). We subsequently grouped these as either polarity defects (“T” or “P” shaped) or formation defects (“N” or “O” shaped). 

Next, we examined animals with the starting extrachromosomal array, *lhEx609*, to determine whether these altered morphologies were caused by the insertion of the transgene. We found that the proportion of morphologies was largely consistent. That is, 73% of animals had a C shape. In total, 10% of animals lacked a visible VD13 commissure (“N” shaped), while a minority of animals had misrouted VD13 axons (“O” shaped). Failure of VD13 to form a commissure or to be misrouted is extremely rare in wild-type *juIs76* animals. Thus, we conclude that the P*plx-2* transgene itself can have a small effect on VD13 development. Overall though, none of the animals we examined had an “H” shaped VD13, which is typical of the other GABAergic MNs. These data argue that the morphology of VD13 could be slightly variable but is principally “C” shaped in wild-type animals. The cell-selective expression of *lhIs97* also suggested that VD13 exhibits characteristics different from the other GABAergic MNs.

### 3.2. VD13 Morphology is Dependent on Wnt-Signaling

Previous work has shown that canonical Wnt signaling regulates the development of posterior GABAergic neurons [29]. In *C. elegans*, there are five genes encoding Wnt ligands (*cwn-1, cwn-2, egl-20, lin-44* and *mom-2*). Of those, *mom-2*, *egl-20*, and *lin-44* are expressed in the posterior of the animal where they are in a position to influence VD13 [30].

We visualized VD13 in animals lacking either *lin-44* or *egl-20*. We found that animals lacking *lin-44* or *egl-20* frequently had VD13 axons that grew past the normal termination point for VD13, similar to previous reports [29]. We also observed VD13 with posteriorly directed neurites (Pdns) emanating from the cell body, confirming these can come from VD13 [31]. Based on these observations we concluded that *lhIs97* was enabling us to confidently observe previously characterized defects in GABAergic neurons.

We then characterized the different neuronal morphologies in the Wnt ligand mutants (Table 1). We found that, in general, when *lin-44* was compared to the wild-type, the same distribution of shapes was found, with no significant differences. The loss of *egl-20* resulted in a significant increase in polarity defects, but not in the formation or completion of commissural outgrowth. 

We next analyzed loss of function mutations in *lin-17/Frizzled, dsh-1/Disheveled* and *mig-5/Disheveled*, and we categorized the penetrance of the different shapes of VD13 neurons in these mutants (Table 1). In the *lin-17, mig-5* or *dsh-*1 mutants, we found a significant increase in the penetrance of dorsal polarity defects, (“T” or “P” shaped neurons), suggesting the polarized growth of neurites in the dorsal nerve cord relied on the function of these proteins. Loss of function in *lin-17* or *dsh-1* had no discernable effect on commissure formation, while there was a significant increase in formation and outgrowth defects in animals lacking *mig-5*. Thus, we confirmed that loss of Wnt signaling genes affected VD13 development, where mutations in ligands versus downstream effectors can be distinct, as has been previously reported [29].

### 3.3. Expression of lhIs97 in VD13 is Dependent on Certain Wnt Pathway Genes

In the course of examining the morphology of VD13 in Wnt signaling mutants, we discovered that mutations in *lin-44, egl-20, lin-17* or *mig-5*, but not *dsh-1*, resulted in an occasional loss of RFP expression in VD13. These animals continued to exhibit expression in the LUA neurons, and VD13 was still present (as labeled by *juIs76*)**, indicating the neurons were still being specified as GABAergic MNs (Figure 3). The penetrance of the *lhIs97* “off in VD13” phenotype was 22% in *lin-44*, <1% in *egl-20*, 30% in *lin-17* and 54% in *mig-5* animals (Figure 3). 

To determine whether *dsh-1* was compensating for the absence of *mig-5*, we analyzed *dsh-1mig-5* double mutants. We found that in this background, *lhIs97* expression in VD13 was lost in 100% of animals analyzed (Figure 3). We next generated *lin-44; egl-20* double mutants, and these animals also exhibited a synergistic decrease, with only ~2% positive for RFP (98% lost expression) (Figure 3). These data indicate that *lin-44* and *egl-20* contributed to VD13 specification in parallel. Finally, we tested the dependence of RFP expression on *bar-1*, which encodes a β-catenin ortholog that functions downstream of *lin-44* in many contexts. We found that *lhIs97* was integrated on the X chromosome, near *bar-1*, and thus, we used the *lhEx610* transgene to examine *bar-1* mutants. Animals lacking *bar-1* also failed to demonstrate expression of the *Pplx-2* transgene in VD13, with 96% losing RFP expression in VD13 (Figure 3). Again, the LUA neurons were grossly unaffected and *juIs76* expression was intact. Overall, these results indicate that the canonical Wnt signaling pathway is critical for VD13 identity, subsequent to GABAergic MN specification.

### 3.4. egl-5 is Necessary for lhIs97 Expression in VD13

The severe perturbation of *lhIs97* expression in the *dsh-1mig-5* and *lin-44; egl-20* doubles led us to look at potential transcriptional targets of the Wnt pathway. Here, Hox genes were obvious candidates. *C. elegans* have an abbreviated Hox cluster with three genes: *lin-39, mab-5* and *egl-5*. Of these, *egl-5* is the most posterior. The *plx-2* promoter was found to contain an EGL-5 binding site by chromatin immunoprecipitation (ChIP) [28] (Figure 1). We found that *egl-5* was expressed in VD13 and the LUA neurons using an EGL-5::GFP translational fusion (*wgIs54)* (Figure 4). We noted that there were many cells that expressed EGL-5 but were not RFP positive in *lhIs97* animals. Thus, *egl-5* is not sufficient to activate RFP expression, but cells that express RFP also express *egl-5*.

We crossed *lhIs97* into a loss of function mutation in *egl-5, n945*, and found that it led to total loss of expression of RFP in VD13 (Figure 5). As with the Wnt mutants, RFP expression was maintained in the LUA neurons and VD13 was still GFP positive by *juIs76* in *egl-5* mutants. Thus, despite the fact that *egl-5* was expressed in the LUAs, it was not essential for expression of RFP in these neurons. 

We attempted to rescue *lhIs97* expression in VD13 by expressing *egl-5* using a GABAergic-specific promoter, *unc-25*. We found that, in addition to recovering RFP in VD13, expression of RFP was ectopically activated in all 19 of the GABAergic MNs (Figure 5). Overall, these results indicate that *egl-5* is both necessary and sufficient for activating the *plx-2* promoter in the GABAergic MNs. However, in non-GABAergic neurons, we do not find that the presence of *egl-5* is sufficient to activate RFP expression from *lhIs97*. Finally, despite the ectopic *egl-5*-dependent expression of RFP in the anterior GABAergic MNs, we did not specifically note that these cells now adopted a C shape. This is not uncommon for Hox transformations, where it is apparently more frequently observed that posterior cells can be transformed to more anterior-like fates by misexpression of Hox genes, but rarely vice versa. 

## 4. Discussion

In *C. elegans*, the loss of a single transcription factor can result in a complete loss of specific behaviors. For example, *ttx-3* results in thermotaxis defects, while the loss of *che-1* results in chemotaxis defects. Subsequent work has found that these transcription factors contribute to the unique identities of individual neurons that regulate these behaviors, which has led to the term “terminal selectors” being applied to these proteins [16,32]. Terminal selectors bind to specific regions of promoters, activating gene expression in only a select set of cells. Part of the goal of understanding the acquisition of neuronal identity is to map terminal selectors to their responsive DNA segments.

Here, we have found a relatively short segment of DNA that is upstream of the *plx-*2/Plexin gene. This 218 bp fragment, when placed upstream of an mCherry reporter gene, results in the expression of RFP in only three of the 302 neurons in the *C. elegans* nervous system. Of these neurons, one is a GABAergic MN, VD13, and two are bilaterally symmetric interneurons (LUAL and LUAR). VD13 and the LUAs are apparently only related by their proximity to one another in the tail of the animal. 

Interestingly, we found that mutations in genes in the Wnt signaling pathway, most specifically, the posterior Hox gene, *egl-5*, resulted in a loss of RFP expression, specifically in VD13, but not the LUAs. Importantly, this loss of identity was not associated with a loss of the GABAergic identity, as these cells still expressed the post-differentiation marker, *unc-25*. And, if we expressed *egl-5* in all GABAergic neurons, we observe robust RFP expression throughout the GABAergic MNs. Animals lacking *egl-5* did not exhibit any alteration in RFP expression in the LUAs, nor was RFP expressed everywhere *egl-5* is found in the animal. This suggests that the DNA segment contains a separate element that is activating expression in the LUAs, dependent on a different terminal selector. Altogether, these results suggest that within VD13, EGL-5 is activating transcription from that DNA segment, suggesting this is a terminal selector and DNA segment pair, and that this was occurring after VD13 was specified as a GABAergic MN. 

This new marker will help us to better understand how GABAergic MNs develop and become individualized during development. Whether VD13 is unique in having another level of specification, or if other DD and VD neurons undergo additional layers of differentiation remains to be seen. In addition, using this promoter, we have shown that this combination of Wnt signaling is instructional for the morphology of VD13. The *lhIs97* reporter illuminates a “C” shaped neuron, versus the sideways H shape typical of the other D-type GABAergic MNs. The lack of anterior processes from the commissure in most of the animals examined suggests that the pattern of axon outgrowth in VD13 is distinct. Loss of *egl-20/*Wnt, *lin-17/*Frizzled and *mig-5* or *dsh-1/*Disheveled resulted in an increase in anterior projections along the dorsal cord. These results suggest that Wnt signaling is normally promoting growth in VD13 toward the posterior direction or inhibiting anterior growth. This is interesting in as much as Wnt ligands have been previously discovered as promoters of GABAergic axon termination in the posterior. This adds yet another layer of complexity in how this signaling pathway might affect neuronal development. The ectopic expression of *egl-5* throughout the GABAergic neurons did not obviously result in the adoption of the VD13 morphology, which suggests that either the morphology is independent of the *egl-5* program, or that anteriorizing factors, which instruct the formation of the H shape, are able to overcome ectopic *egl-5* expression.

This is not the only context where *egl-5* functions in this manner. Notably, *egl-5* functions in the touch receptor neurons (TRNs) to induce further morphological changes to define the PLM neuron, which differentiate them from the standard touch receptor AML neuron [33]. We do not see expression of our *egl-5-*dependent RFP reporter in the touch neurons, suggesting that EGL-5 is regulating this via a different DNA contextual element in the TRNs. Thus, Wnt signaling and *egl-5* are used as terminal selectors in multiple neuronal developmental contexts to differentiate neurons within established classes (e.g., mechanosensory, GABAergic, etc*.)*.

Going forward, it is important to continue to define the interactions of the Wnt signaling pathway and *egl-5* in their role as terminal selectors. Namely, our results are in agreement with previous work showing that the Wnt pathway can have some contradictory results when analyzing neural development. With regard to the expression of *lhIs97* in VD13, loss of the ligands and effectors appear to act equivalently. Conversely, with regard to the morphology of VD13, loss of the ligands is somewhat discrete when compared to the effectors. The consequences on morphology are consistent with other observations that Wnt ligand mutations result in axon termination errors, specifically observed as axon overgrowth. When analyzed, *lin-17/*frizzled receptor mutants had both under and overgrowth, and the disheveled adaptors, as well as the β-catenin *bar-1*, are exclusively undergrown [29]. In our hands, loss of Wnt ligands had only a modest effect on dorsal process polarity, while *lin-17, mig-5* and *dsh-1* had very strong effects. Within the specification of VD13, *lin-44; egl-20* double mutants were stronger than loss of *lin-17* alone, suggesting there are other Wnt receptors involved in this process. However, those double mutants were the same as removing *bar-*1, suggesting that *bar-1* is working downstream of both *lin-44* and *egl-20* in this event. Wnt signaling may then, somehow, be activating *bar-1* during early events (VD13 specification) but may function via a different mechanism during axon termination. Further defining this relationship and exploring factors downstream of *egl-5* will allow us to better define the manner in which these terminal selectors function to modulate gene expression during neural development.

Overall though, our observations suggest that Wnt signaling is contributing to many different aspects of neuronal development for these neurons. The ability to use a cell-selective marker to interrogate those functions will enable new mechanistic questions to be asked. Unfortunately, due to the fact that loss of function in some Wnt signaling genes renders our single cell marker unusable, we cannot examine the morphology of VD13 alone in *egl-5* mutants or the *dsh-1mig-5* or *lin-44; egl-20* double mutants. However, knowing that there is an additional layer of specification that occurs in VD13 may enable us to find other markers that may be transcriptionally independent of Wnt signaling and permit us to visualize VD13 in those backgrounds. 

## Figures and Tables

**Figure 1 jdb-08-00005-f001:**
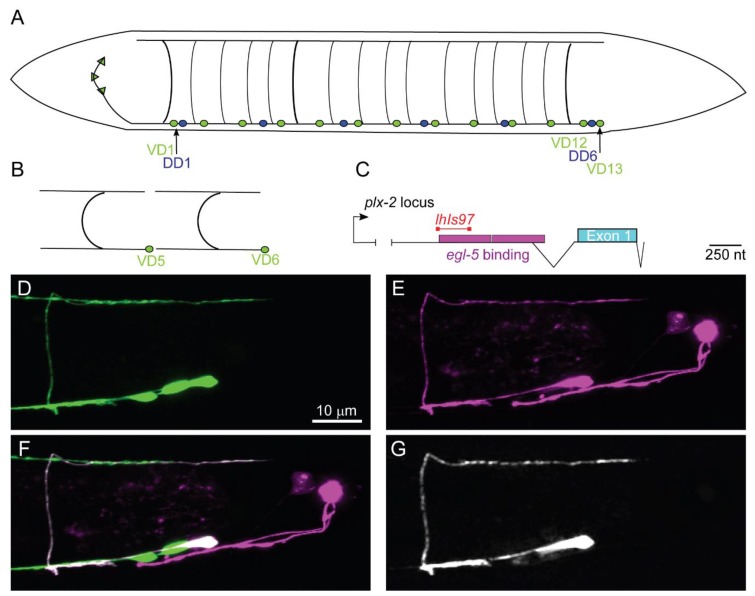
*lhIs97* is a VD13-selective GABAergic motorneuron (MN) marker. (**A**) A schematic of the 19 GABAergic MNs. DD1 and VD1 are the most anterior and VD13 is the most posterior. (**B**) The characteristic tiled H shape of the VD neurons is illustrated. (**C**) A schematic of the *plx-2* locus. The region upstream of the *plx-2* first exon contains two *egl-5* binding sites [28]. RNASeq experiments have identified an intron in the 5′ region (indicated by the branched line). The segment of the promoter used to generate the *lhIs97* marker is indicated in red. (**D**–**G**) The posterior region of an *lhIs97* wild-type animal showing GABAergic GFP (**D**), *lhIs97-*expressed RFP (**E**), the merge of the two channels (**F**) and the overlap of the GFP and RFP channels, which is exclusive to VD13 (**G**).

**Figure 2 jdb-08-00005-f002:**
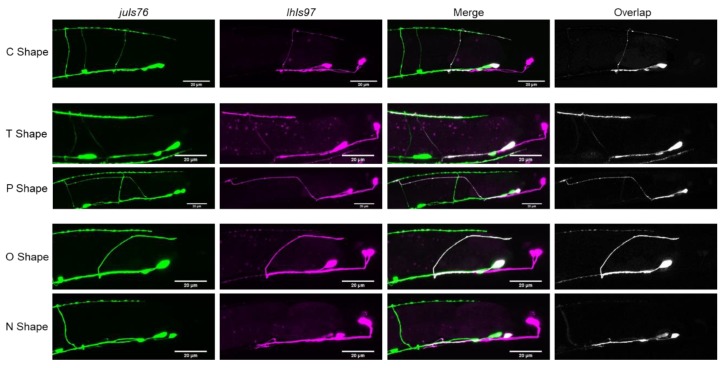
Different VD13 morphologies observed. The most common shape of VD13 in *juIs76; lhIs97* animals was a C shape. Aberrant morphology categories observed were either in the dorsal cord polarity (“T” or “P” shaped) or commissural outgrowth and guidance (“O” or “N” shaped). Note, the “C”, “T”, and “O” shaped morphology examples are wild-type animals, “N” shaped, *dsh-1* mutants, and “P” shaped from *egl-20* mutants.

**Figure 3 jdb-08-00005-f003:**
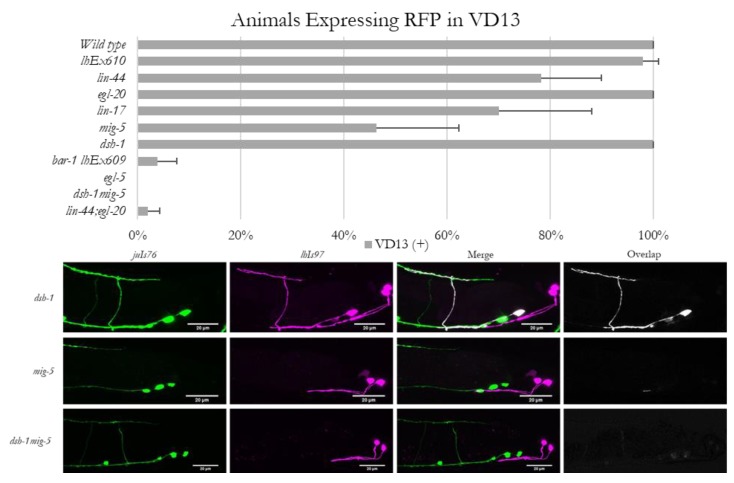
Expression of *lhIs97* is selectively lost in VD13 in Wnt loss of function animals. Animals were scored for the presence or absence of RFP in VD13. The percentage of animals (mean ± s.d.) that expressed RFP in each genetic background is presented in the graph. Below are examples of RFP expression being present in VD13 in *dsh-1* mutants, but absent in the majority of *mig-5* single mutants and all *dsh-1mig-5* double mutants.

**Figure 4 jdb-08-00005-f004:**
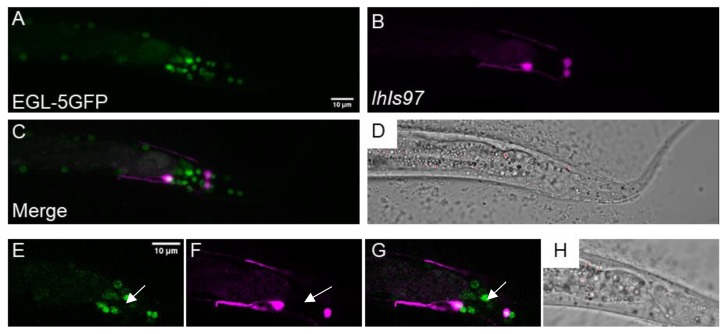
*egl-5* is expressed in VD13 and the LUA neurons. (**A**) EGL-5::GFP, under the control of the *egl-5* promoter, was observed in the posterior of the animal. (**B**) *lhIs97* expresses RFP in VD13 and the LUAs. (**C**) The signal is co-incident in both cells, suggesting that *egl-5* is expressed in both VD13 and the LUAs. (**D**) A transmitted light image of the region. (**A**–**D**) A z-projection through the entire animal. (**E**–**H**) A single plane image through the center of VD13. The arrow indicates VD13.

**Figure 5 jdb-08-00005-f005:**
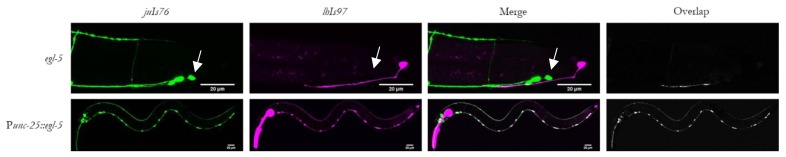
*egl-5* is necessary and sufficient for *lhIs97* expression in GABAergic neurons. Top row, *egl-5(n945)* mutants have lost RFP in VD13 (arrow). Bottom row, ectopic expression of *egl-5* in all GABAergic neurons results in RFP expression in these cells.

**Table 1 jdb-08-00005-t001:** VD13 morphology by genotype.

Genotype	N	C Shape	Polarity (T/P)	Outgrowth (N/O)
wild type (*lhIs97*)	161	82%	9%	9%
*lin-44(n1792)*	88	77%	18%	5%
			(*P* = 0.0446)	(*P* = 0.3135)
*egl-20(gk453010)*	159	64%	23%	14%
			(*P* = 0.003)	(*P* = 0.0769)
*lin-17(n671)*	40	48%	50%	3%
			(*P* < 0.0001)	(*P* = 0.6961)
*mig-5(rh97)*	85	36%	52%	12%
			(*P* < 0.0001)	(*P* = 0.0339)
*dsh-1(ok1445)*	177	24%	71%	5%
			(*P* < 0.0001)	(*P* = 0.2113)

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
