# Peer review of "The Hox Gene egl-5 Acts as a Terminal Selector for VD13 Development via Wnt Signaling"

_jdb, 2020, doi:10.3390/jdb8010005_

Round 1
Reviewer 1 Report
The manuscript by Kurland et al. explores transcriptional programs controlling the development of morphological and functional features that define specific neuronal classes amongst C. elegansGABAergic motor neurons. Using a newly developed reporter incorporating a 218 bp promoter fragment of the plx-2gene to distinguish the most posterior of the VD GABAergic neurons, VD13, the authors demonstrate that wild type VD13 processes do not extend anteriorly past the commissure, and tease apart Wnt signaling pathways that affect VD13 morphological features, likely through regulation of the hoxtranscription factor egl-5. The authors show that combined mutations in mig-5and dsh-1, 2 of the 3 worm dissheveled genes, strongly alter VD13 morphology and expression of the plx-2reporter. Likewise, mutation of egl-5eliminates expression of the VD13 reporter. The paper describes a useful reagent and uses this tool to uncover a transcriptional pathway that contributes to the unique morphological features of VD13. I have the following specific recommendations for strengthening the manuscript.
1. Page 4, lines 142-3: the authors could determine if the higher rate of morphological defects in lhIs97 animals compared to juIs76 animals is a consequence of the integrated transgene by examining other integrated lines or non-integrated transgenics.
2. Page 5: it is not clear if the morphological defects in lin-44mutants were deemed significant or not. The text states that the effects were not significant (lines 162-3), but then later states there is a significant increase in anterior processes (line 169).
3. Did the authors test egl-20;lin-44double mutants?
4. The effects of the dsh-1 mig-5double mutant are stronger than observed for mutation of lin-17/Frizzled. Might this indicate contribution of a parallel Wnt pathway that is engaged independently of lin-17? The authors should comment.
5. Although not essential for the manuscript as written, data indicating whether Wnt signaling pathway mutations also alter egl-5::GFPexpression would strengthen the links between Wnt signaling and egl-5transcriptional regulation. Similarly, it would be interesting to know whether mutation of egl-5alters VD13 morphology?
6. The arrows in Figure 5 appear misplaced.
Author Response
- Page 4, lines 142-3: the authors could determine if the higher rate of morphological defects in lhIs97 animals compared to juIs76 animals is a consequence of the integrated transgene by examining other integrated lines or non-integrated transgenics.
This is a good idea. We have done this experiment and included it in the revised version.
- Page 5: it is not clear if the morphological defects in lin-44mutants were deemed significant or not. The text states that the effects were not significant (lines 162-3), but then later states there is a significant increase in anterior processes (line 169).
There were no significant differences in the percentage of animals in the different morphological categories (after a multiple-test correction). However, we confirmed using this marker that VD13 exhibits aberrant termination of growth in lin-44 animals, which has been reported in depth in other articles, we do apologize for the confusion. The process referred to on line 169 of the previous version is the posterior process, there is no statistically significant difference in the number of lin-44 animals where VD13 form an anterior process. This was not true for egl-20, where we did find differences to be statistically significant. We have amended the text to be clearer about the phenotypes.
- Did the authors test egl-20; lin-44double mutants?
That is also a good idea. We have, and those data are now included in the manuscript.
- The effects of the dsh-1 mig-5double mutant are stronger than observed for mutation of lin-17/Frizzled. Might this indicate contribution of a parallel Wnt pathway that is engaged independently of lin-17? The authors should comment.
That is also a good point, and, as with the double mutants of lin-44 and egl-20, there are certainly combinations of frizzled mutants we can investigate, although with 4 different frizzled genes, the number of combinations becomes more complicated. There are also other potential Wnt receptors, cam-1, etc. Thus, we feel that the lin-44; egl-20 and dsh-1mig-5 double mutants provide confidence that there is compensation between Wnt signaling proteins. The exact role for each component will require additional observations. We have amended the manuscript per the reviewer’s suggestions to point out that the data likely indicate lin-17-independent events, either from an additional frizzled, or possible non-frizzled Wnt receptors that we have yet to identify.
- Although not essential for the manuscript as written, data indicating whether Wnt signaling pathway mutations also alter egl-5::GFP expression would strengthen the links between Wnt signaling and egl-5 transcriptional regulation. Similarly, it would be interesting to know whether mutation of egl-5alters VD13 morphology?
We agree. We have a preprint (Hartin et al., on bioRxiv) that demonstrates that bar-1/beta-catenin is required for egl-5 expression in VD13. That manuscript is currently under review, and thus, the data cannot be included here. Please note, that manuscript is focused on the role of syndecan in axon outgrowth as part of the Wnt pathway, dependent on egl-5. There are no data in that paper describing the differentiation of VD13 as a function of egl-5, nor are we repeating the morphological characterization of VD13 in Wnt mutants. Rather, we will cite this manuscript to support those conclusions.
We would very much like to analyze VD13 morphology in egl-5 mutants, but our marker is 100% off in these animals, and thus it is not possible at this time to do so. We failed to point out specifically that in egl-5 mutants the axons stop short of the normal termination point (visible in Figure 5). So there is a consequence of loss of egl-5 on VD13 outgrowth, but how the morphology is altered is very much an open question right now. We can add that, in the ectopic expression of egl-5 we do not seem to see conversion of the anterior H-shape neurons into C-shapes, suggesting that this is not sufficient to drive a new morphology. However, this is consistent with other Hox gene transformations, where it is more difficult to convert anterior cells to more posterior fates. Thus, we cannot rule out that in the egl-5 loss of function VD13 may adopt a new morphology, one that is more similar to the anterior VD cells.
- The arrows in Figure 5 appear misplaced.
We thank the reviewer, and have made the corrections.
Reviewer 2 Report
In this work Kurland et al, describe a new marker for VD13 neurons (the product of a region upstream of plx2 gene) in C. Elegans, they also show how depletion of Wnt pathway signalling components reduce the expression of this marker. Moreover, they show that Egl5, a HOX gene is necessary for such marker to be expressed and that overexpression of Egl5 is sufficient to induce the expression in the rest of GABA-ergic motoneurons. In conclusion, the paper describes an enhancer DNA sequence that is expressed exclusively in the VD13 motoneurons.
Main concerns:
1- Although the association of Wnt signalling pathway and GABAergic neurons is well demonstrated, it is not really a novelty, as it had been mostly reported before.
2- Egl5 is with no doubt mediating the activity of the described enhancer, however in my opinion it is not proved that Egl5 is working downstream of Wnt pathway in this situation.
3- It would be interesting to know what protein product is produced in response to the discovered enhancer because it would unravel what pathway is primary activated to define the identity of the VD13 neurons.
4- I am not really convinced that Egl5 is a terminal selector for VD13, and not just one of many transcription factors that participate in the differentiation of these cells.
Minor, but not less important concerns:
In my opinion, the paper is chaotically written and it is difficult to learn which the objectives of the work are.
Author Response
In this work Kurland et al, describe a new marker for VD13 neurons (the product of a region upstream of plx2 gene) in C. Elegans, they also show how depletion of Wnt pathway signaling components reduce the expression of this marker. Moreover, they show that Egl5, a HOX gene is necessary for such marker to be expressed and that overexpression of Egl5 is sufficient to induce the expression in the rest of GABA-ergic motoneurons. In conclusion, the paper describes an enhancer DNA sequence that is expressed exclusively in the VD13 motoneurons.
Main concerns:
1- Although the association of Wnt signaling pathway and GABAergic neurons is well demonstrated, it is not really a novelty, as it had been mostly reported before.
We understand and have cited the previous work on Wnt signaling and GABAergic development. The role of Wnt signaling on GABAergic differentiation has not been reported previously.
2- Egl5 is with no doubt mediating the activity of the described enhancer, however in my opinion it is not proved that Egl5 is working downstream of Wnt pathway in this situation.
We appreciate the reviewers concern. One issue is that any Wnt – egl-5 double mutant would be difficult to interpret given that egl-5 mutants have a fully penetrant phenotype. We have found that the bar-1 phenotype is equivalent to egl-5, and provided evidence in other contexts where this link is consistent, for other phenotypes. Thus, we are proposing egl-5 is functioning downstream of Wnt, in this context, however, it is genetically difficult to determine for sure.
3- It would be interesting to know what protein product is produced in response to the discovered enhancer because it would unravel what pathway is primary activated to define the identity of the VD13 neurons.
The enhancer is upstream of the plx-2 gene, and therefore most likely regulates the expression of the Plexin-2 protein, a receptor for semaphorin, a known axon guidance cue. We obtained a plx-2 mutant, but did not observe significant differences in VD13 growth. We are attempting to make a double mutant with the second Plexin, to determine if they function redundantly. However, at this point, it is unlikely that plx-2 is important for VD13 specification, or that this pathway is only regulating plx-2 expression. We have conducted a genetic screen to identify other molecules that affect the activity of this enhancer, and have isolated 8 new alleles, but the identity of the genes affected is not yet known.
4- I am not really convinced that Egl5 is a terminal selector for VD13, and not just one of many transcription factors that participate in the differentiation of these cells.
To be honest, I am not sure that there is a difference. A 2016 paper by Oliver Hobert (Terminal Selectors of Neuronal Identity) lays out the concept. Many of the genes listed as terminal selectors, for example, unc-30, participate in the differentiation of the cells. In fact, I believe that is part of the explicit definition, with the additional idea that the transcription factor plays a role in the final steps in the specification. In the same paper, he lays out an approach of using fluorescent reporters that are active in a subset of cells and identifying singular or combinatorial cases in which expression is selectively disrupted. Here we show that egl-5 loss of function results specifically in the loss of expression of a marker in VD13, and that ectopic expression activates this marker in other, competent, cells. Thus, I would be interested to understand what parts of the terminal selector concept we have failed to demonstrate.
Minor, but not less important concerns:
In my opinion, the paper is chaotically written and it is difficult to learn which the objectives of the work are.
We have attempted to make the manuscript more accessible to the non-expert. However, if there are specific places the reviewer finds difficult to read, please let us know.
Round 2
Reviewer 2 Report
The paper is now ready to be published.